# Muscular Strategies for Correcting the Pelvic Position to Improve Posture—An Exploratory Study

**DOI:** 10.3390/jfmk9010025

**Published:** 2024-01-29

**Authors:** Oliver Ludwig, Carlo Dindorf, Sebastian Kelm, Jens Kelm, Michael Fröhlich

**Affiliations:** 1Department of Sport Science, Rheinland-Pfälzische Technische Universität Kaiserslautern-Landau (RPTU), 67663 Kaiserslautern, Germany; carlo.dindorf@rptu.de (C.D.); skelm@rptu.de (S.K.); michael.froehlich@rptu.de (M.F.); 2Orthopädisch-Chirurgisches Zentrum, 66557 Illingen, Germany; jk66421@hotmail.de

**Keywords:** postural weakness, posture correction, pelvic tilt, muscle activation, muscular imbalance, clustering

## Abstract

The correction of postural weaknesses through the better positioning of the pelvis is an important approach in sports therapy and physiotherapy. The pelvic position in the sagittal plane is largely dependent on the muscular balance of the ventral and dorsal muscle groups. The aim of this exploratory study was to examine whether healthy persons use similar muscular activation patterns to correct their pelvic position or whether there are different motor strategies. The following muscles were recorded in 41 persons using surface electromyography (EMG): M. trapezius pars ascendens, M. erector spinae pars lumbalis, M. gluteus maximus, M. biceps femoris, M. rectus abdominis, and M. obliquus externus. The participants performed 10 voluntary pelvic movements (retroversion of the pelvis). The anterior pelvic tilt was measured videographically via marker points on the anterior and posterior superior iliac spine. The EMG data were further processed and normalized to the maximum voluntary contraction. A linear regression analysis was conducted to assess the relationship between changes in the pelvic tilt and muscle activities. Subsequently, a Ward clustering analysis was applied to detect potential muscle activation patterns. The differences between the clusters and the pelvic tilt were examined using ANOVA. Cluster analysis revealed the presence of four clusters with different muscle activation patterns in which the abdominal muscles and dorsal muscle groups were differently involved. However, the gluteus maximus muscle was involved in every activation pattern. It also had the strongest correlation with the changes in pelvic tilt. Different individual muscle patterns are used by different persons to correct pelvic posture, with the gluteus maximus muscle apparently playing the most important role. This can be important for therapy, as different muscle strategies should be trained depending on the individually preferred motor patterns.

## 1. Introduction

Poor posture in the sagittal plane, such as a hollow back or a hunchback, can lead to long-lasting pain, which is now a major economic factor in healthcare worldwide. Incidences of 71% have been reported in adulthood [1] and 25 to 60% in childhood and adolescence [2,3,4,5].

Posture itself is a multifactorial phenomenon and, in addition to sociocultural factors, is also significantly influenced by muscular balance, strength, and flexibility. The basis for the alignment of the spine is the human pelvis with the sacrum anchored in it, whose position in the sagittal plane is influenced by the balance of the forces of different muscle groups [6]. The classical mechanical model assumes that the gluteus maximus and hamstrings lead to a lowering of the posterior and the lifting of the anterior pelvic edge (also called retroversion) when activated and, thus, to a reduction in the pelvic tilt. On the other hand, the activation of the straight and oblique abdominal muscles (rectus abdominis, obliquus, and transversus muscles) can initiate a synergistic movement if the position of the sternum is fixed at the same time. The iliopsoas muscle, the rectus femoris muscle, and the lumbar back muscles (erector spinae muscle) act as counterparts to this retroversion of the pelvis, causing an increased forward tilt (the so-called anteversion) of the pelvis [6]. The pelvic position at rest therefore depends on the equilibrium of the forces of the activated muscles, assuming in this simplified mechanical model that the muscles on both sides of the body are always activated symmetrically (Figure 1).

This approach already shows that several muscle groups can work synergistically and that the same movement goal (i.e., lifting the front edge of the pelvis) is made possible by redundant muscle activity.

Numerous studies in recent years have supported this model. For example, Kim et al. [7] found that the lordosis angle of the lumbar spine is less influenced by the absolute forces of the trunk extensors and flexors but rather by the force ratio of both muscle groups. An imbalance between the two antagonistic muscle groups leads to a significant change in the lumbar lordosis and has been identified as a possible risk factor for low back pain. Bridger et al. [8] emphasized the influence of the length and stretchability of the iliopsoas muscle on pelvic position, as well as the importance of the hamstrings for positioning the pelvis. These results were confirmed by Jorgensson et al. [9], who found a correlation (albeit weak; r = 0.4033, *p* = 0.001) between the iliopsoas muscle length and the lordosis, with the lordosis increasing with muscle shortening.

Klee et al. [10] were able to show that people with strongly developed abdominal muscles exhibit a lower anteversion of the pelvis, while people with strongly developed hip flexors show a more forward-tilted pelvis. The importance of the abdominal muscles for pelvic erection was also emphasized by Hodges et al. [11], who attributed an important role to the transversus abdominis muscle in stabilizing the lumbo-pelvic region. The M. multifidii and M. erector spinae have, in turn, been identified as agonists of an increased pelvic tilt [12].

An important approach to clarifying the influence of different muscle groups on pelvic position is intervention studies that examine the changes in posture in the course of stretching and strengthening or fatigue programs. For example, López-Miñarro et al. [13] showed that the static stretching of the hamstrings led to an immediate change in the pelvic position (in the sense of a forward tilt) and sagittal spinal curvature. Preece et al. [14] were able to show the same for the hip flexors, the stretching of which led to an immediate reduction in the anterior pelvic tilt. Yoo [15] achieved an increase in the anterior pelvic tilt by training the ventrally downwards pulling muscles in a patient with hypolordosis of the lumbar spine. Alvim et al. [16] found an increase in the pelvic tilt with fatigue of the extensor portion of the M. gluteus maximus. Fundamental work was carried out by Klee [10], who was able to prove the influence of the aforementioned muscle groups on pelvic tilt through extensive intervention programs, and was thus able to refute the findings of Levine et al. [17], who found no influence of the abdominal muscles on pelvic tilt.

Due to the lifestyle of modern civilization, the pelvic erector muscles weaken, for example, due to long periods of sedentary work and a lack of physical balance, while the pelvic tilting muscles shorten. In this context, Czaprowski et al. [18] emphasized that, when planning corrective exercises, the hyperactivity and hypoactivity of these muscles must be considered in addition to examining their shortening status.

If the pelvic tilt then increases due to a muscular imbalance, the sacrum tilts forward with the pelvis. As a result, the base of the fifth lumbar vertebra also shifts ventrally and a compensatory increased hyperlordosis of the lumbar spine develops [17,19]. Physiotherapy for a tilted pelvis therefore requires a complex approach. In addition to compensating for muscular deficits (strengthening and stretching), training the sensorimotor processes (perception of the pelvic position and conscious activation of the relevant muscles) also leads to an improvement in the pelvic position and, therefore, often forms the basis for correcting the posture [20].

Targeted strengthening, stretching, and body awareness exercises as part of sports therapy or physiotherapy are effective against the postural deficits caused by an increased pelvic tilt. Any therapeutic treatment should not only be evidence-based but also (cost-) efficient. This means, for example, that forms of exercise must be used in a very targeted manner in order to achieve the greatest possible benefit for the patient in the time available. In this context, Barczyk-Pawelec et al. [21] emphasized the need to use individually suitable physiotherapy measures for the patient in the form of targeted exercise programs.

According to current biomechanical models, strengthening the muscle chain that pulls the pelvis upwards ventrally (rectus abdominis muscle and obliquus muscle) should, therefore, be just as effective as training the muscles that work dorsally towards the caudal direction (gluteus maximus muscle and biceps femoris muscle). These synergistically working muscles form a redundant system; different muscle activations can theoretically achieve the same biomechanical effects (here: retroversion of the pelvis).

Muscle synergies are like building blocks that consist of characteristic activation patterns across multiple muscles that may be different for each person but fulfill similar functions [22]. Such muscle synergies have been investigated for various holding and movement tasks, for example, for lower limb movement [23]. In the context of redundant systems, they also represent an important control mechanism for the central nervous system [24]. In order to understand basic muscular synergies, it is important to first work these out in healthy test persons, as it is known that muscular activation patterns can change in the presence of complaints [25].

Against the background of this muscular redundancy, however, it has not yet been clarified whether people use the same muscular strategies to consciously straighten the pelvis, i.e., activate the muscles mentioned in the same way, or whether the muscles that are activated voluntarily to correct the pelvis differ from person to person. This is of particular therapeutic interest, as the effect of differentiated training exercises may well be different for different people since the muscles that are maximally strengthened are not necessarily also used by the central nervous system for pelvic correction. In these cases, a therapy that only activates individual muscles may not have the desired effect.

The aim of the present exploratory study was, therefore, to clarify whether healthy persons activate different synergistically working muscles for voluntary pelvic straightening or whether similar activation patterns can be found.

## 2. Materials and Methods

A total of 41 male participants (age 21.65 ± 3.50 years, height 178.98 ± 3.28 cm, weight 77.10 ± 6.09 kg, BMI 24.05 ± 1.49 kg/m^2^) took part in the study. The test persons were recruited from sports clubs and among university students. In order to form a homogeneous group of participants, only men were included in this study. As a sedentary lifestyle is discussed as a possible cause of low back pain [26] and at the same time back pain can alter muscular activation [27], only persons who did moderate exercise (more than 3 h per week) and had an active, nonsedentary lifestyle were included. The exclusion criteria were acute orthopedic or internal diseases or complaints, as well as leg length differences greater than 5 mm. All participants were informed about the procedure before the start of the study and gave their written informed consent.

### 2.1. Preparation

The participants stood in front of a calibration wall in their underwear. Foam marker spheres with a diameter of 10 mm were attached to the ASIS (anterior superior iliac spine) and PSIS (posterior superior iliac spine) on the left side of the body using double-sided adhesive tape. A horizontally aligned camera on a tripod (Panasonic HC-V777, Full HD, Panasonic Corp., Kadoma, Japan) filmed the pelvis from the side.

Before the start of the test, the skin in the area of the muscle groups under examination was prepared. The skin preparation was carried out in accordance with the SENIAM standards [28]. If necessary, the hair was removed with a razor, the skin was cleaned with alcohol, and the blood circulation was stimulated locally by rubbing vigorously with a cloth. Pre-gelled electrodes (Ambu BlueSensor P, Ag/AgCl, diameter 38 mm, sensor area 10 mm^2^, Ambu GmbH, Bad Nauheim, Germany) were attached to the skin over the following six muscle groups on the left side of the body: M. trapezius pars ascendens, M. erector spinae pars lumbalis, M. gluteus maximus, M. biceps femoris, M. rectus abdominis, and M. obliquus externus. The electrodes were placed 40 mm (center point to center point) apart after palpation of the corresponding anatomical landmarks, if possible in the middle of the muscle bellies [29]. The adhesive electrodes were connected to the transmitter via approx. 7.6 cm long cables. These were fixed to the prepared skin at the same distance from both electrodes using double-sided adhesive tape and had an in-built reference electrode. The cables were routed in a loop so that no tensile stress due to skin displacement occurred on the electrodes during movement. The quality of the electrophysiological recording was checked during voluntary contraction of the recorded muscle. The positioning of the electrodes is also shown in Figure 2.

One problem when analyzing surface EMG data is that a good signal-to-noise ratio must be ensured when evaluating the signals and that electrode displacements can contaminate the raw signal [23]. In order to counter these problems, the noise was reduced by good skin preparation, for example, and the signal-to-noise ratio was checked before the analysis. According to the manufacturer (Noraxon Inc., Scottsdale, AZ, USA), the amplifier has a common-mode rejection ratio (CMRR) > 100 db and a signal quality < 1 µV RMS baseline noise. Artefacts caused by skin displacements during the simple movement investigated were not visible due to correct electrode and transmitter fixation.

### 2.2. Measurement Procedure

The participants stood sideways to the camera in a resting position, looking straight ahead, with their arms crossed in front of their chest. At the experimenter’s command, the participants straightened their pelvis ten times, which means that they performed a posterior pelvic tilt (retroversion of the pelvis). The instruction was standardized and consisted of the command to raise the front edge of the pelvis for one second and then slowly lower it again. If the test persons had difficulty visualizing the movement, the instructor told them to imagine the pelvis as a wheel with its axis in the hip joints and then to turn this wheel in the direction indicated. All participants were able to perform the movement correctly, which could be verified by measuring the pelvic tilt. Between the individual repetitions, the test persons were asked to remain in the resting position for 30 s. The resting position consisted of an upright relaxed posture in which the test persons were asked to relax their muscles. The arms remained crossed in front of the chest even in the resting position. This position could also be monitored via the video recording.

The electrode signals were transmitted telemetrically to the EMG system (Noraxon TeleMyo DTS, Noraxon, Scottsdale, AZ, USA) and recorded at 1000 Hz. The system operates with an input range of ±6.3 mV and has a 16-bit analog-to-digital converter with dynamic resolution. The EMG system allows telemetric data transmission of up to 30 m; in our tests, the distance between the transmitters and the receiver was only 4 m. Isometric maximum strength tests (MVC) were then performed once for each muscle group and the EMG activity was measured during these tests. Depending on the muscle group tested, the participant laid in the prone or supine position on an examination couch. Two other experimenters fixed the trunk or extremities (with the hip and knee joints in the neutral zero position), while the test person pressed against them isometrically with maximum force for 5 s. To avoid fatigue, the maximum strength tests were only carried out once.

### 2.3. Data Processing

The telemetrically recorded data were further processed using the program Myo Muscle 3.12.56 (Noraxon, Scottsdale, AZ, USA). First, a band-pass filter (10–300 Hz) was applied. No notch filters (50/60 Hz) were used. After rectification, RMS (root mean square, 100 ms window) was applied to smooth the signal. The onset behavior of the muscle was determined based on the threshold by the point of maximum slope of the envelope (maximum of the 1st derivative). From this point onwards, the muscle onset was defined and additionally verified by visual inspection by an expert, and the rectified raw signal was averaged [30]. Since we only evaluated the amplitude level, the determination of the muscle onset was not as time-critical as in studies that require a determination of this parameter in the lower millisecond range. Afterwards, the averaged EMG amplitude was calculated for each of the ten repetitions for the time interval of one second starting with the specified muscle onset by adding the rectified signals and dividing by the time. Finally, these 10 values were averaged and then normalized as a percentage of the MVC for each subject. To determine the MVC, the raw data from the isometric maximal strength tests were filtered as described above, rectified, and averaged over the 5 s intervals.

The angle between the horizontal and the connecting lines of the markers on the ASIS and PSIS (the so-called pelvic tilt) was calculated in the resting phase and at the time of maximum pelvic straightening in the video freeze frame using the Dartfish Pro Suite vers. 6 software (Dartfish, Fribourg, Switzerland). The amplitude of pelvic movement (pelvic tilt delta) was calculated and averaged over all 10 subtests. The reliability of measuring pelvic tilt using videographic methods and the suitability of this parameter for assessing the pelvic position have been well researched and the scientific quality criteria have been confirmed in multiple studies [31,32,33,34].

### 2.4. Statistics

A linear regression analysis was conducted to assess the relationship between the pelvic tilt delta and the muscle activities. Outlier detection was previously performed, ensuring that none of the values exceeded three times the standard deviation. Additionally, an in-depth examination of the studentized deleted residuals, leverage values, and Cook’s distances was carried out, resulting in the exclusion of data from three persons who were identified as outliers after a further expert evaluation. All necessary assumptions, including normality of residuals, homoscedasticity, linearity, and independence of errors, were thoroughly tested using diagnostic plots, Shapiro–Wilk tests for normality, scatterplots of residuals versus predicted values, and Durbin–Watson tests. These tests confirmed that the assumptions for the regression analysis were valid. The coefficient of determination (R^2^) and the adjusted counterpart (adjusted R^2^) are given to evaluate the goodness of fit and are interpreted according to Cohen [35].

Based on the standardized maximum muscle activities, significant variables were identified through both stepwise and backward regression analyses. Subsequently, a clustering analysis was applied to the variables representing muscle activities to unveil potential patterns or activation strategies. Ward clustering was employed for this purpose, and prior to clustering, the data underwent standardization using z-scores.

The evaluation of the number of clusters was started with a comprehensive analysis using objective metrics. This involved silhouette score analysis using scikit-learn [36] and dendrogram techniques, with SciPy [37] used to automatically determine the number of clusters. The default cut-off value was set to 0.7 times the maximum distance between clusters.

The silhouette score analysis involved iteratively evaluating cluster solutions for different numbers of clusters and calculating the average silhouette score for each configuration. The optimal number of clusters was then determined by selecting the configuration with the highest silhouette score. Both the silhouette score analysis and the dendrogram techniques yielded consistent results, highlighting the robustness of the clustering solution. In addition, the identified solution was validated and checked for plausibility by a subsequent expert-based evaluation as described in Forina et al. [38].

Following clustering, differences between the clusters in terms of pelvic tilt were examined using ANOVA. To address multiple comparison issues, a Bonferroni alpha error correction was applied, and the significance level was set at 0.05. The necessary test requirements were checked and could be assumed. All the calculations were executed in IBM SPSS Statistics (Vers. 29.0.0, IBM Corp., Armonk, NY, USA). The visualizations were generated using Seaborn [39] and Scipy [37].

## 3. Results

The correlations of the variables are presented in Figure 3. The pelvic tilt delta was correlated with the muscle activations of the gluteus maximus and lower trapezius with correlation coefficients of 0.35 and 0.25, indicating a weak correlation according to Cohen [35]. The lower trapezius and erector spinae lumbalis were correlated with each other (correlation coefficient 0.41), followed by the rectus abdominis and obliques (correlation coefficient 0.37).

The stepwise and backward regression analyses yielded consistent results. The models only included the gluteus activity as an independent variable. The gluteus activity had a significant effect on the pelvic tilt delta (F (1, 37) = 6.915, *p* = 0.01). The R^2^ for the overall model was 0.16 (adjusted R^2^ = 0.14), which is indicative of a moderate goodness-of-fit according to Cohen [35]. Figure 4 visualizes the relationship between the variables.

The cluster analysis revealed the presence of four clusters according to the dendrogram displayed in Figure 5. A silhouette score of 0.22 suggested a moderate level of cohesion and separation in the clustering [40]. Figure 6 compares the variables per cluster. Cluster A was characterized by strong activity of the abdominal muscles and the gluteus maximus. Cluster B showed strong activation of the gluteus and back muscles. Clusters C and D showed hardly any activity in the abdominal and back muscles but strong activity in the gluteus maximus. In Cluster C, the hamstrings were also strongly active. The ANOVA results showed that the pelvic tilt delta did not significantly differ between the clusters (F (3, 34) = 0.84, *p* = 0.48).

## 4. Discussion

The correction of the pelvic position is an important therapeutic approach in the treatment of typical postural weaknesses such as an increased pelvic tilt or hyperlordosis of the lumbar spine (hollow back). This requires the strengthening of weak postural muscles in order to restore an altered muscular balance [6]. Raising the front edge of the pelvis leads to a steeper position of the sacrum due to the backward rotation of the pelvis, thereby reducing lumbar lordosis [19,41], and is, therefore, an important posture-constituting feature. Since the human muscle system is redundant, many movements can be realized by different muscle activation patterns. The aim of our study was to find out whether there exist certain uniform, subject-independent muscular activation patterns that contribute to pelvic straightening.

The investigation of muscular synergies is the subject of intense neurophysiological research. Many movement tasks can be realized through different activation patterns due to the redundant function of muscle groups [42,43,44]. This has been proven in various studies, e.g., for the lower limbs [45,46,47,48]. Such patterns consist of muscle activities that are coordinated in terms of time and amplitude and are switched together. Synergistic interaction between different muscles has several advantages. Firstly, a synergist can provide support when a muscle is energetically exhausted. This is particularly important for the pelvic muscles, as they have to guarantee the stability of the pelvic position even when standing for long periods and thus create the basis for a stable spinal column position. Secondly, greater forces can be generated by the co-contraction of multiple muscles, which may be necessary when lifting heavy loads, for example.

From a biomechanical perspective, the lifting of the anterior pelvic edge could occur via two different muscle groups: via the ventral group, in particular, the rectus abdominis and obliquus externus and internus, or via the dorsal group, gluteus maximus, and hamstrings [6,8] (see Figure 1). The iliopsoas muscle, rectus femoris muscle, and lumbar erector spinae muscle act antagonistically in terms of an increase in pelvic tilt [15].

In our study, the gluteus maximus, which moves the posterior upper pelvic edge caudally and thus leads to the elevation of the anterior upper pelvic edge (retroversion), had the strongest correlation with the pelvic tilt. This is consistent with the studies by Alvim et al. [16], who were able to show that the forward tilt of the pelvis increases when the gluteus maximus muscle is fatigued. Nevertheless, it must be critically noted that we only found a moderate correlation according to Cohen [35], which may indicate redundant muscle activity.

The cluster analysis showed a total of four clusters (Figure 5), in which the combinations of muscles activated to straighten the pelvis differed (Figure 6). However, the clusters did not differ in terms of the degree of pelvic movement initiated (pelvic tilt delta), i.e., they were all equally efficient. This study showed that different people use different muscle activity patterns to correct their pelvic position. Figure 7 shows these patterns schematically.

Cluster A shows strong activation of the straight and oblique abdominal muscles with the involvement of the dorsal muscles, especially the gluteus. In this pattern, the synergistic effect of the muscles working ventrally to cranial and dorsally to caudal becomes particularly clear. In this context, Levine et al. [49] were unable to show that strengthening the abdominal muscles caused a reduction in the pelvic tilt. However, it must be borne in mind that an improvement in individual strength components (maximum strength or strength endurance) may not necessarily lead to a change in the joint position, as activation by the CNS is a key factor. Even a maximally strong muscle would be ineffective for pelvic correction if it is not activated in a targeted manner by the CNS as part of a motor strategy. However, the studies by Barczyk-Pawelec et al. [21] support our findings, as they were able to show that adolescents with a weak posture also had weaker abdominal trunk muscles.

On the other hand, Cluster C showed a clear dominance of the dorsal muscles inserting at the pelvis. In this case, the biceps femoris works synergistically with the gluteus and moves the posterior edge of the pelvis downwards. While in Cluster C, the gluteus maximus and the biceps femoris were equally activated, in Cluster D, pelvic straightening was achieved almost exclusively via the gluteus maximus, which made it a central muscle in the correction.

It is interesting to note the differentiation from Cluster B, which primarily activated the back muscles (trapezius and erector spinae) in addition to the gluteal muscles. This does not seem plausible at first, as the lumbar erector spinae tends to cause the pelvis to tilt forward [12]. The increased activation of the upper back muscles in particular would be understandable against the biomechanical background that the sternum must also be fixed as an anchor point when using the abdominal muscles for pelvic straightening. However, we were unable to demonstrate this. During the strongest activation of the ventral muscle group (Cluster A), the superficial back muscles were only moderately active. In this respect, we can only assume that this was possibly a nonfunctional activation if the participants were not familiar with the required pelvic straightening movement sequence. Nevertheless, the test persons who showed this muscular activity were able to correct the pelvis just as well as all other groups. It should be noted at this point that only superficial muscle potentials were recorded in this exploratory study. The activation of deeper muscle groups that stabilize the spine was, therefore, not measurable. However, it can be assumed that these muscle groups are constantly active in an upright posture.

Even though such different patterns were found in our study, they all contain the gluteus maximus muscle as a common component. One explanation for this may be the high proportion of force that can be generated by this muscle [50]. It must also be considered that the given task of pelvic correction requires voluntary activation of the muscles, whereby the gluteus can be voluntarily contracted without any problems, but the abdominal muscles, for example, may be more difficult to control in terms of posture correction [51]. The fact that the gluteus maximus plays a central role in all the patterns may be important in the preventive or therapeutic correction of poor posture, as the intervention could be more targeted if familiar activation patterns are used and may, therefore, be more effective.

The study design of our exploratory study is subject to some limitations. For example, it must be emphasized that the state of stretching or shortening of the muscles also has an influence on the habitual pelvic position, but this was not recorded in our study. For example, López-Miñarro et al. [13] found that static stretching of the hamstrings increased the anterior pelvic tilt. Cejudo et al. [52] were also able to prove the relationship between hamstring extensibility and pelvic mobility. Jorgensson et al. [9] were able to show that the lumbar lordosis increased with increasing shortening of the iliopsoas muscle. This was confirmed by Preece et al. [14], who were able to show that stretching the hip flexor muscles can lead to an immediate reduction in the pelvic tilt during a habitual stance.

It must also be borne in mind that we worked with healthy test persons, meaning that the results cannot be directly transferred to patients. Furthermore, only adult men with moderate physical activity took part in our study. It is therefore not possible to directly transfer the results to minors, women, or people with a predominantly sedentary lifestyle. As we did not measure the maximum forces of the muscle groups examined, no conclusions can be drawn about the possible (maximum) strength deficits. Further research should show whether the different activation clusters are possibly related to strength deficits in individual muscle groups. In addition, further studies with a larger sample and more electrophysiologically registered muscles would be useful in order to possibly be able to further differentiate the clusters.

Nevertheless, we consider the results to be relevant for therapy, as muscle strength training will presumably have the greatest effect on pelvic posture if the patient knows how to use the trained muscles to correct posture. This requires the voluntary activation of individual muscle groups, which would mean using previously learned patterns rather than learning new patterns. This consideration could also explain the negative results of Levine et al. mentioned above [49], as their approach of strengthening the abdominal muscles to improve the pelvic tilt possibly only worked in persons who knew or used this muscular strategy.

This means, for example, that patients who primarily correct their posture using the activation patterns from Cluster D may benefit less from training their abdominal muscles than from training their gluteus maximus. Patients who activate their muscles according to Cluster A, on the other hand, may benefit from training their abdominal muscles.

For the trainer or therapist who does not know and cannot measure the activation strategies of a person, this means that varied training of the dorsal and ventral muscle groups promises the greatest success. Since the gluteus was activated in all clusters and can be controlled very well by the test persons at the same time, its training should be emphasized. At the same time, it must be borne in mind that improving sensorimotor functions improves the success of pure strength training for posture correction [20] and that practicing corrective movements can, therefore, also change motor strategies.

## 5. Conclusions

In healthy persons, muscular patterns were identified in which muscles work synergistically to correct the pelvic position in the sense of retroversion. In one pattern, the abdominal muscles and gluteus maximus work together, in another, the gluteus and biceps femoris, and in a third, the gluteus maximus is almost exclusively responsible for pelvic straightening. In another cluster, nonspecific, nonfunctional activity of the back muscles was also measured. Each of the patterns found produced an equally effective pelvic correction. By using the individual activation patterns in a therapeutic pelvic correction, a more efficient therapy can be expected. In any case, special attention should be paid to the training of the gluteus maximus muscle, as it is represented in each of the four patterns.

## Figures and Tables

**Figure 1 jfmk-09-00025-f001:**
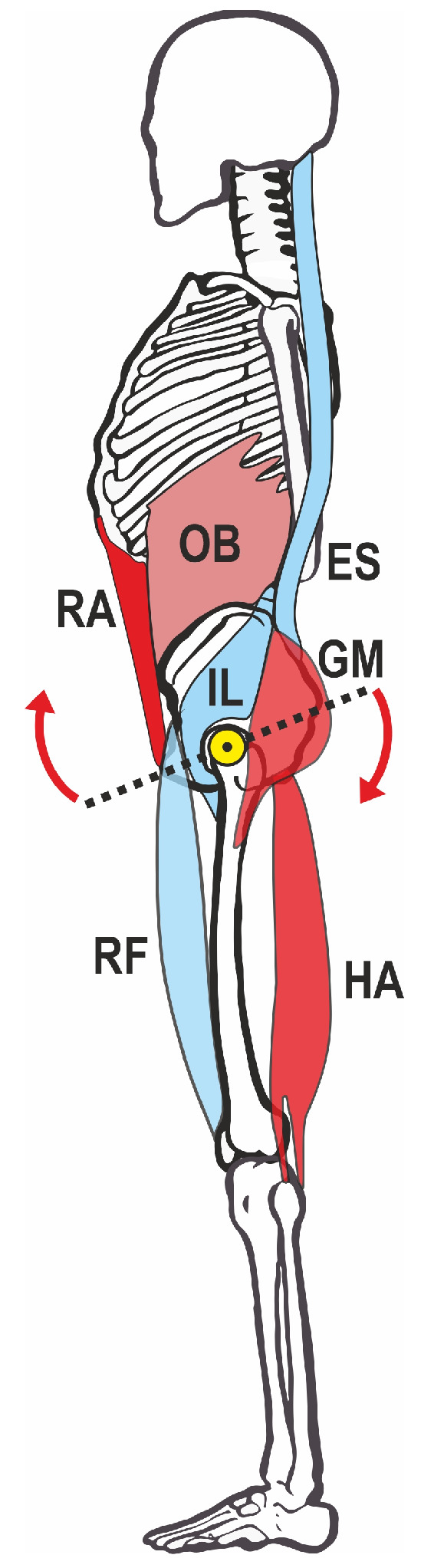
Mechanical model of pelvic straightening. Red: muscles that lift the anterior part of the pelvis, blue: antagonists. The yellow dot marks the pivot point around the hip joint, arrows mark the direction of anteversion movement. HA—hamstrings, OB—obliquus, ES—erector spinae, GM—gluteus maximus, RA—rectus abdominis, IL—iliopsoas, RF—rectus femoris.

**Figure 2 jfmk-09-00025-f002:**
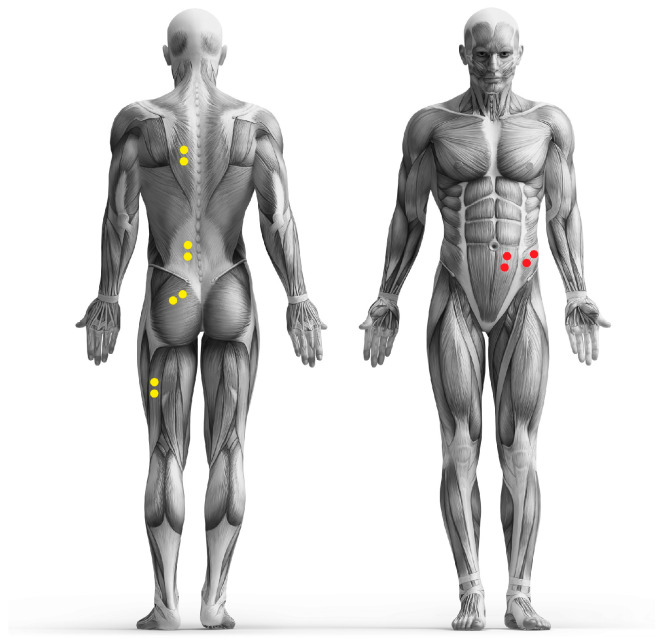
Placement of the electrodes (yellow and red dots).

**Figure 3 jfmk-09-00025-f003:**
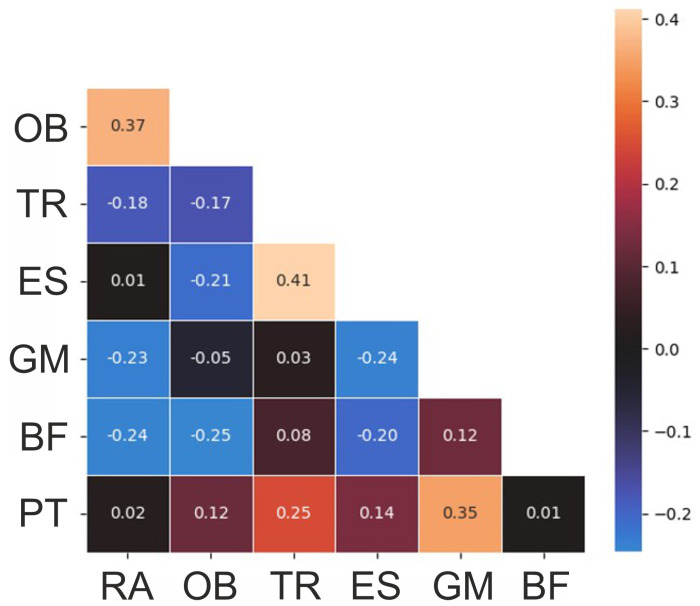
Correlation heatmap of the variables. RA—rectus abdominis, OB—obliquus, TR—trapezius pars ascendens, ES—erector spinae pars lumbalis, GM—gluteus maximus, BF—biceps femoris, PT—pelvic tilt delta.

**Figure 4 jfmk-09-00025-f004:**
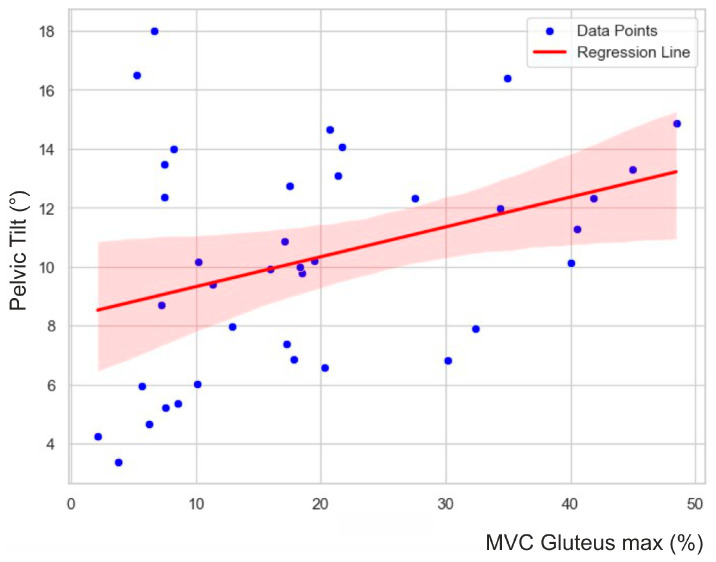
Scatter plot with regression line, summarizing the linear relationship and associated uncertainty. Blue points represent individual data observations, while the red line indicates the positive correlation between the gluteus maximus and pelvic tilt delta. The shaded area around the line represents the 95% confidence interval, representing the likely range of the true regression line.

**Figure 5 jfmk-09-00025-f005:**
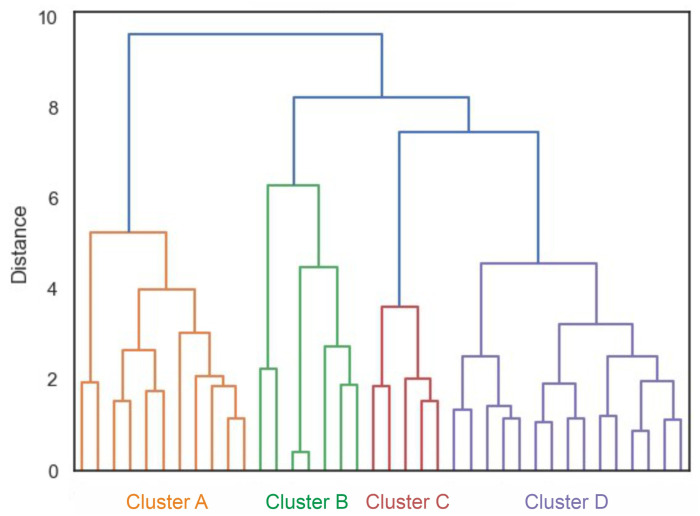
Dendrogram of the cluster analysis. Different colors mark the different clusters.

**Figure 6 jfmk-09-00025-f006:**
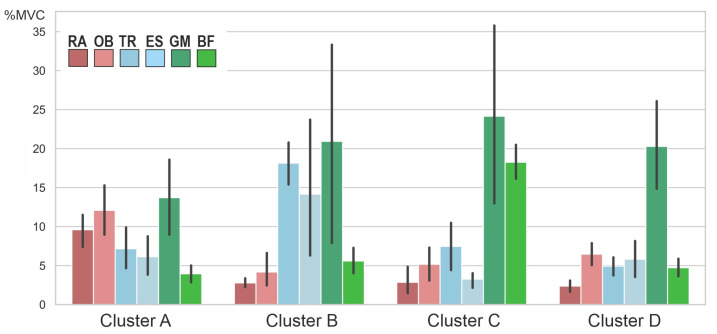
Averaged muscle activity (in % MVC) within the four clusters. RA—rectus abdominis, OB—obliquus, TR—trapezius pars ascendens, ES—erector spinae pars lumbalis, GM—gluteus maximus, BF—biceps femoris. Error bars represent 95% confidence intervals. Note that the nonstandardized values are displayed for better interpretation; the z-standardized values were used for the cluster analysis itself.

**Figure 7 jfmk-09-00025-f007:**
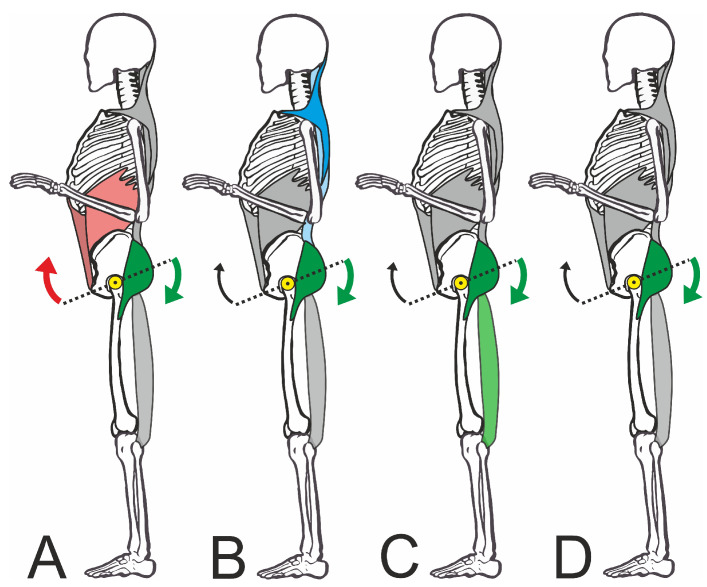
Schematic representation of the four muscle activation patterns. The subfigures (**A**–**D**) are explained in detail in the text. The colored muscles are primarily activated. Arrows: retroversion of the pelvis; yellow dot: pivot point of the pelvis in the hip joint.

## Data Availability

The data will be made available upon justified request.

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
