# Peer review of "Muscular Strategies for Correcting the Pelvic Position to Improve Posture—An Exploratory Study"

_jfmk, 2024, doi:10.3390/jfmk9010025_

Round 1

Reviewer 1 Report

Comments and Suggestions for Authors

This study explores subtle differences in muscle activation patterns during voluntary pelvic straightening and their potential impact on postural correction. However, certain key aspects need to be clarified and highlighted for a more complete understanding. The primary role of the gluteus maximus and the relationship of overall muscle balance in affecting pelvic alignment should be clearly and specifically explained. The introduction should emphasize the importance of studying healthy individuals and list exclusion criteria for subject selection. In the materials and methods section, the criteria for healthy subjects must be clearly defined, detailing the specific steps of the experiment. The results and discussion section should provide a detailed analysis of the results of the study and acknowledge the inherent limitations of excluding children and adolescents. This conclusion will help elucidate the role of different muscle activation patterns in correcting pelvic position, underscoring their importance in addressing pelvic tilt problems in healthy subjects. Overall, refining clarity, providing specific criteria, and emphasizing practical implications will significantly improve the relevance of research to postural therapy.

Specific comments are shown below:

Abstract:

1. The appeal of a summary is essential to guide the reader further into the content of the article. Given the ambiguity of the results and conclusions in the present abstract, it is suggested to refine and clarify.

2. Lines 10-11: Previous studies have shown that pelvic position in the sagittal plane is influenced by muscle balance in the ventral and back muscle groups, while the conclusion highlights the primary role of the gluteus maximus in this process. There may be some confusion here. It is necessary to clarify the relationship between the two and how to highlight the role of the gluteus maximus in the conclusion.

Introduction:

1. The introduction should emphasize the significance of studying healthy individuals in exploring whether they activate different synergistic muscles for voluntary pelvic straightening and whether similar activation patterns can be identified.

2. Lines 126-128: “These synergistically working muscles form a redundant system…”. For the interpretation of muscle synergy patterns in different work situations, this needs to be supported by relevant studies (The Effect of Pelvic Floor Muscle Training on Pelvic Floor Dysfunction in Pregnant and Postpartum Women; Adaptive Neuro-Fuzzy Inference System model driven by the Non-Negative Matrix Factorization-extracted muscle synergy patterns to estimate lower limb joint movements).

Materials and Methods:

1. Lines 141-142: The authors do not clearly define the criterion of "healthy person" here, which is crucial in the study. The introduction mentions that prolonged sitting and lack of exercise can lead to pelvic tilt, but it is not clear whether the subjects ruled out chronic sedentary and lack of exercise habits. It is recommended that the authors clearly describe the criteria for subject selection, in particular, whether individuals who are chronically sedentary and inactive are excluded.

2. Lines 155-159: For the specific location of the electrode connection, it is recommended that the author provide a more precise and clear description so that the reader can accurately understand the position of the electrode. This makes it easier for the reader to reproduce the experiment or refer to relevant information.

3. Line 161: Regarding resting positions, ask the author to describe in detail the specific positions the subject took during the experiment. A clear description will help the reader to understand the experiment accurately.

4. Line 163: Regarding pelvic straightening, the author is asked to provide a more specific description, including how to guide the subject through the process of pelvic straightening and the state of the pelvis after straightening. This will give the reader a clearer understanding of the subject's movement performance and state changes in the experiment.

5. Lines 175-186: The authors are requested to provide a more detailed description of the processing of EMG signal data, as well as muscle activation solving. This is crucial, both in the credibility of the results and the reader's understanding of the article. The authors can refer to the following studies, which are described in detail in their methods section: Accurately and effectively predict the ACL force: Utilizing biomechanical landing pattern before and after-fatigue; Adaptive Neuro-Fuzzy Inference System model driven by the Non-Negative Matrix Factorization-extracted muscle synergy patterns to estimate lower limb joint movements. For example, the recursive model (second-order differential equation) was conducted to solve the muscle activation by the obtained normalized signal. Please revise.

Result:

1. The results section is insufficient to summarize the main findings and significance of the study. It is recommended that the authors provide a more definitive summary of the main conclusions of the study, with particular emphasis on understanding the effect of different muscle activity patterns on pelvic alignment in healthy people, and the potential value of this finding for postural correction therapy. This is crucial because, for the treatment of postural problems, understanding the differences in specific muscle activity patterns may be crucial for the personalized selection of treatment approaches. Therefore, highlighting this point can further highlight the importance of the research findings and the potential for clinical application.

Discussion:

1. The study was designed to explore whether different synergistic muscle groups are activated in healthy subjects when performing voluntary pelvic straightening, and whether similar activation patterns exist, but unfortunately, this goal is not fully discussed in the discussion section.

2. In addition, it is mentioned in the introduction that the incidence of pelvic tilt is higher in children and adolescents, but this study was only tested on adult men. Therefore, it is recommended to take this into account when discussing the limitations of the study.

Conclusion:

1. Line 423: "Different people correct pelvic positions with different patterns of muscle activation" requires a more specific explanation and elaboration of the classification results. This conclusion is too general and more detailed information is needed, such as about the specific types of different muscle activation patterns, how these patterns differ from each other, and how they affect pelvic position adjustment.

2. In addition, the role of muscle activation patterns in healthy subjects in correcting pelvic tilt is not mentioned in the conclusion, which is an important point worth emphasizing. It is suggested that the importance of activating different muscle patterns in healthy people to correct pelvic tilt should be added to the conclusions to highlight the practical significance of the findings.

Comments on the Quality of English Language

no

Reviewer 2 Report

Comments and Suggestions for Authors

Well organized and written manuscript. Some clarifications are needed. See pdf for specific comments.

Round 2

Reviewer 1 Report

Comments and Suggestions for Authors

With some necessary modifications, the quality of Manuscript has improved significantly. The reviewer has the following minor concerns:

As the reviewer mentioned in the previous comment: “The authors are requested to provide a more detailed description of the processing of EMG signal data, as well as muscle activation solving”. The authors have made some modifications to this, however the relevant details should be strengthened. The authors may consider referring to the two studies mentioned by the reviewers in the previous comment, which could help improve the robustness of the methodology.

Comments on the Quality of English Language

no
